# Laser Reconstruction of Spinal Discs Experiments and Clinic

Andrei Baskov [1,2,3], Igor A. Borshchenko [3], Vladimir Baskov [1,3], Anatoly Shekhter [4,5] and Emil Sobol [6,7,*]

1   Russian Medical Academy of Continuous Professional Education of the Ministry of Healthcare Department of Neurosurgery, 125993 Moscow, Russia; abaskov@mail.ru (A.B.); baskov_v@mail.ru (V.B.)
2   Department for Spine Surgery, Central Clinical Hospital of Russian Railways, 121359 Moscow, Russia
3   Orthospine Clinic, 125252 Moscow, Russia; spine@orthospine.ru
4   Department of Experimental Morphology and Biobanking, Institute for Regenerative Medicine, Sechenov First Moscow State Medical University (Sechenov University), 119991 Moscow, Russia; a.shehter@yandex.ru
5   World-Class Research Center "Digital Biodesign and Personalized Healthcare", Sechenov First Moscow State Medical University (Sechenov University), 119991 Moscow, Russia
6   Beckman Laser Institute & Medical Clinic, University of California, Irvine, CA 92697, USA
7   Arcuo Medical Inc., Incline Village, NV 89450, USA
*   Correspondence: esobol@uci.edu

**Featured Application: Laser tissue modification in orthopedics and reconstructive medicine.**

**Abstract:** Degenerative disease of the intervertebral discs (DDD) is currently a serious problem facing the world community. The surgical methods and conservative therapy used today, unfortunately, do not stop the pathological process, but serve as a palliative method that temporarily relieves pain and improves the patient's quality of life. Therefore, at present, there is an active search for new methods of treating DDD. Among new techniques of treatment, biological methods, and minimally invasive surgery, including the use of laser radiation, which, depending on the laser parameters, can cause ablative or modifying effects on the disc tissue, have acquired considerable interest. Here, we analyze a new approach to solving the DDD problem: laser tissue modification. This review of publications is focused on the studies of the physicochemical foundations and clinical applications of a new method of laser reconstruction of intervertebral discs. Thermomechanical action of laser radiation modifies tissue and leads to its regeneration as well as to a long-term restoration of disc functions, elimination of pain and the return of patients to normal life.

**Keywords:** laser modification; tissue structure; regeneration; cartilage repair; spine; intervertebral disc; degenerative disc diseases

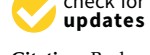

## 1. Introduction. Degenerative Diseases of the Spine. Causes, and Modern Methods of Treatment

### 1.1. Anatomical, Morphological, and Clinical Concepts of Degenerative Diseases of the Spine

Degenerative diseases of the intervertebral discs (DDD) are currently a serious problem facing the world community [1,2]. The prevalence of DDD is roughly described in proportion to age, so that 40% of people in their 40 s have DDD, rising to 80% in people aged 80 and over. Each year, 403 million new cases of DDD are registered [3].

The intervertebral disc (IVD) consists of three main parts: the annulus fibrosus (AF), the nucleus pulposus (NP) and the dense hyaline endplate (EP), which is in the intimate system of the vertebral plate closure (Figure 1A). The AF surrounds the NP and consists of dense plates, each of which is formed by longitudinal and parallel collagen fibers (CF), and the direction of these fibers in the adjacent plates is mutually perpendicular. AF consists of fibrous cartilage in which CF are type I and type II collagens. Proteoglycan (PG) complexes—aggreganes—are contained between CFs. Aggreganes are composed of hyaluronic acid and a core protein to which chains of glycosaminoglycans (GAGs) are attached, mainly chondroitin and keratan sulfates [4].

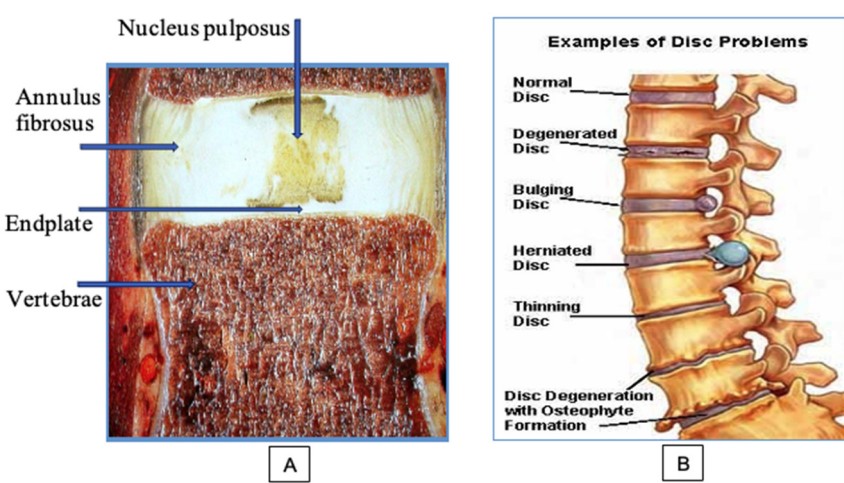

**Figure 1.** Structure and problems of the spinal disc: (**A**) main components of the disc; (**B**) the sequence of the development of degenerative disc diseases.

The NP is the inner part of the IVD; it consists of a highly hydrated tissue of a jelly-like consistency. The basis of the NP is formed by water (80–90%) and large compounds of GAG. The sulfate groups of GAGs carry negative charges that contribute to the maintenance of osmotic pressure in the disc, which ensures the amortization functions of the spine. The amount of cells in the IVD is very low (0.25–0.5%). Cells must maintain the macromolecular structure of the disc, a certain concentration of PG, collagen, proteases, and their inhibitors throughout life. Inside the disc of an adult, blood vessels are completely absent, and nutrition of the cartilaginous structures is carried out from the bodies of the vertebra by diffusion and osmosis through the EP. IVD remains healthy as long as the balance between the synthesis and decay of its macromolecules is maintained.

At the initial stages of development of DDD, the NP is gradually dehydrated. A decrease in pressure inside the disc produced pathological mobility in the vertebral segment. Clinically, this is manifested by the appearance of local moderate pain in the damaged area, a feeling of fatigue, and recurrent attacks of myogenic pain syndrome. Changes appear in MRI. Losing moisture, NP, normally light and homogeneous, gradually fragments and becomes dark in color. This phenomenon is called a dark disc. The progression of the degenerative process in the IVD, dehydration of the NP, and an increase in segment instability led to a significant increase in the load on the AF. As a result, the number of AF microcracks increases; these microcracks gradually transform into functionally significant cracks and ruptures (Figure 1B). Fragments of the NP begin to penetrate these cracks. With sharp physical exertion, the process can be aggravated to a complete rupture of the AF with the loss of sequestered fragments of the destroyed NP into the spinal column.

### 1.2. Modern Methods of Treatment of Degenerative Disc Diseases

Treatment of DDD is usually based on surgical methods or conservative therapy, including medical pain relief and the use of physiotherapy methods of treatment. Conservative therapy, used for several years, leads to a decrease in pain, but also leads to the appearance of IVD fibrosis. At the same time, the persisting overload of the facet joints of the spine and the phenomena of instability form a new pain syndrome [5]. The use of modern surgical technologies for the treatment of DDD, unfortunately, does not stop the pathological process, but serves as a palliative method that temporarily relieves pain and temporarily improves the patient's quality of life. Surgical interventions lead to the subsequent decompensation of the process and weaken the compensatory capabilities of the organism itself. Therefore, at present, there is an active search for new methods of treating DDD, and colossal funds are spent on research in this area [6]. Thermal inactivating pain receptors in IVD is carried out by percutaneous radiofrequency (RF) thermo-coagulation or intradiscal electro thermotherapy (IDET) [7–10]. RF and IDET methods do not stimulate

biological disc remodeling and repair. After some time, the pain syndrome progresses again [11–13]. Various methods of nucleotomy have become another area of intradiscal decompression [14]. Some authors suggested that fenestration and partial removal of IVD material would reduce intradiscal pressure and nerve root irritation. For this, both endoscopic and puncture access were used, with various types of nucleotomes under X-ray control. Despite the low invasiveness, cost and comparative safety of puncture techniques, the problem of relapses, postoperative instability, and low efficiency (60–70%) of such operations remains. During long-term follow-up from 5 to 25 years for patients after open classical discectomy, based on radiological changes, acceleration of disc degeneration was noted in 48.7% of cases. Microsurgical technique reduces this risk, provoking negative changes in 9.1% [15]. One of the problems, especially in the case of operations aimed at removing a hernia, is a recurrent herniated disc, which can reach 10–15% of operated cases [16,17]. The implants used to close the defects of the AF designed to reduce the number of relapses did not confirm their effectiveness in the long-term period after surgery [18].

At the beginning of the 21st century, fusion and implantation operations became widespread [19]. However, there are randomized studies that show that the placement of implants increases the risk of undesirable effects (nerve damage, blood loss, inflammatory and general complications), while increasing the time of surgery and the risk of revision spine surgeries [20,21].

In recent years, attempts to counteract degenerative disc damage is also associated with biological methods with intradiscal stem cell transplantation. Despite some laboratory advances, the clinical use of stem cell transplantation has led to mixed results. In particular, after transplantation of one's own mesenchymal stem cells into degeneratively affected IVDs with characteristic pain syndrome in provocative discography, after a year, none of the patients showed a decrease in the intensity of back pain [22]. At the same time, some researchers have found beneficial effects in reducing pain and improving disc morphology [23,24]. Four to six years after such procedures, their safety is reported; however, disc changes were not of a unidirectional, favorable nature, as they combined both positive and negative signs of the progression of the degenerative process [25].

*1.3. Lasers in Spine Surgery*

Many studies on the use of lasers in orthopedics refers to ablative destructive surgery of the spine. For the first time, percutaneous laser discectomy (PLD) was performed in 1987 using a Nd: YAG laser [26]. With the development of the PLD technique, its clinical success increased to 80% or more [27]. It was found that in the tissues surrounding the ablation zone significant damage is possible due to heating. This can cause complications, often including an infectious lesion of the operated IVD—discitis in the zone of tissue necrosis after laser evaporation of the central part of the NP. There is aseptic discitis, root damage, thermal damage to the EPs of adjacent vertebrae. One year after PLD, necrotic tissues are found in the disc. A decrease in the likelihood of side effects has been achieved through selective ablation of the NP and by reducing the radiation power. Transforaminal epiduroscopic laser ablation (TELA) selectively removed the herniated portion of IVD and induced more effective decompression with minimal complications [28]. TELA enhances the resorption of the herniated IVD by drilling a hole in the AF and partially removing the slipped NP with the use of a 1414 nm Nd:YAG laser. Although the clinical success rate was 76.6% at 12 months, it is unclear whether a better outcome can be achieved with direct AF ablation.

In [29], a technique for laser irradiation of an IVD with a moderate power—thermodiscoplasty (TDP)—is presented, which is based on the effect of collagen compression when heated above 70 °C. TDP leads to tissue compaction and reduction in its volume without carbonization or significant destruction. The TDP method is used during endoscopic removal of a hernia or disc protrusion, when using an endoscope, it is possible to accurately bring the laser probe into the area of the pain generator, including after mechanical removal

of the restrained fragments of the NP. At the same time, in the zone of laser exposure, there was histologically a thickening and homogenization of CFs as well as destruction of most of the cells. Laser radiation made it possible to vaporize the NP of the disc, while simultaneously producing its de-reception. The success rate was 95.2%, and in many cases, the postoperative control MRI showed a decrease in protrusion size. Even though the total energy of laser action on the IVD was significantly lower (2–3 times) than with the accepted puncture laser discectomy, this action nevertheless caused extensive tissue coagulation and, consequently, the destruction of cellular elements. This, unfortunately, significantly worsened the dynamics of reparative processes in the IVD.

A fundamentally new approach to the treatment of DDD is based on laser-induced modification of the IVD tissues [30]. Moderate heating and control of thermomechanical stresses in tissues subjected to short-term non-destructive pulse repetitive laser irradiation make it possible to form new cartilaginous tissue of the fibrous-hyaline and hyaline types in the IVD [31]. Activation of reparative regeneration was observed as early as on the 7th day, and replacement of AF and NP defects with new cartilaginous tissue occurred 2–3 months after laser exposure. The developed technology of laser reconstruction of discs (LRD) makes it possible to treat IVD by replacing disc disrupture and disc fissures with regenerating cartilaginous tissue [32,33]. This contributes to the reverse development of the degenerative-dystrophic process with stabilization of the dorsal segment due to the replacement of altered IVD tissues with regenerating hyaline and fibro hyaline cartilage [30–33]. In addition, the formation of a new dense cartilage limits the migration of NP fragments, preventing the recurrence of disc herniation [34]. This is due to the transformation of the NP into hyaline cartilage, which tightly grows together with neighboring tissues and without clear boundaries passes into AF and hyaline EP. The thermal effect of laser radiation leads to de-reception of the disc and the effect of "compression", which ensures a decrease in the size of the protrusion. Moreover, the replacement of degeneratively destroyed IVD areas with new regenerating cartilaginous tissue limits the expansion of the innervation zone in the disc and thereby eliminates the development of discogenic reflex pain syndromes [33–35].

Currently, among neurosurgeons and orthopedists, there are two opposing points of view on the possibilities, advantages, and disadvantages of ablative laser technology for the treatment of IVD. Along with a critical but generally positive assessment of the results of laser surgeries, the technology of which goes through various stages of development, a skeptical assessment of the prospects for the use of ablative lasers in spinal surgery has received a certain spread. The authors of [36] write: "Despite the fact that lasers have long been used in pain treatment procedures such as percutaneous discectomy, the alleged benefits of lasers, such as reducing inflammation and degeneration, have not been supported by reliable clinical studies." On the other hand, the authors of [37] note that after a 2-year follow-up, the strategy of laser percutaneous disk decompression by results is not inferior to microdiscetomy and, as a minimally invasive method, can take its place in the arsenal of treatment for radiculitis caused by herniated IVD. An example of a negative assessment of ablative laser surgery is the work [38], which notes that several randomized clinical trials have shown the lack of clinical efficacy of PLD-type procedures compared to microdiscectomy. It is noted that laser procedures are often performed by pain specialists without special surgical experience who are unable to eliminate the postoperative complications of PLD. There are doubts about the feasibility of using PLD for the treatment of minimal disc herniation, since such patient problems often resolve spontaneously within a few months without surgery. In a recent review article on the use of lasers in neurosurgery [39], the following conclusion was formulated: "The many ways in which lasers are used in neurosurgery are evidence of the technological advances and practicality of laser science. Despite the difference of opinion, the use of lasers for minimally invasive procedures shows promising results and deserves further study."

It should be emphasized that the above doubts relate mainly to destructive operations on the IVD using ablative laser radiation. The authors of negative statements did not consider the new restorative technology of LRD, which eliminates the adverse consequences

associated with damage to surrounding tissues [31–35]. In contrast to ablation technologies, the use of moderate radiation powers in LRD allows non-destructive modification of the tissue microstructure. For the processes of laser modification, the correct choice of parameters is especially important, which makes it possible to control the ongoing physicochemical and biological processes by controlling the temperature and thermomechanical stresses in the laser treated zone. This makes it possible to obtain a stable positive effect of LRD [33,35].

## 2. Non-Destructive Laser Modification of Cartilage Is the Basis of the New LRD Approach

### 2.1. Physicochemical and Biological Processes in the Disc Tissues under Laser Radiation

When exposed to laser radiation on disc tissue, various processes occur, including optical (absorption and scattering of light), thermal (heating and cooling), mechanical (the formation and propagation of mechanical stresses, changes in the strength characteristics of the tissues), electrical (redistribution of charges, change in electrical conductivity) and physicochemical processes (rupture of chemical bonds, phase transitions and chemical reactions, formation of gas bubbles, denaturation, coagulation, tissue carbonization). At the same time, various biological processes (proliferation and death of cells, transformation of the cartilage matrix, and tissue regeneration) also occur, which can lead to positive or negative medical effects.

Laser-induced phase transformations and chemical reactions in tissues occur in several stages. The first stage of laser modification of the structure and mechanical properties of cartilage is a change in the state and structure of interstitial water. Depending on the power and duration of laser heating, the subsequent stages of laser exposure can be various processes, such as structural changes in the collagen and proteoglycan subsystems, local mineralization, local melting, and the formation of micropores in the cartilage matrix [40]. The study of collagen stability in AF was carried out by the method of differential scanning calorimetry [41]. It was shown that the violation of the structural organization of the IVD collagen network is determined not only by the maximal temperature of the heated zone, but also by the space–time temperature distribution. Short-term (within 5–10 s) heating of IVD tissues to temperatures of 65–70 °C does not lead to a noticeable denaturation of CF. At the same time, the non-uniform temperature distribution leads to the expansion of heated zones, the movement of interstitial water and the development of thermal stresses in the affected area. As a result, local foci of violation of the strictly ordered arrangement of CFs and fibrils arise. If the temperature of the tissue in this case turns out to be above 80 °C, then a noticeable denaturation of collagen occurs, which leads to local shrinkage of the CFs. This process, in turn, stimulates the development of additional tension on adjacent fibers, which can lead to rupture of individual CFs]. Thermomechanical stresses affect cells, the structure and thermal stability of the extracellular matrix (ECM), both in the heat-affected zone and outside it [41]. The results of these experiments can explain the reasons why various currently used physical methods associated with significant heating of AF (including laser, electrothermal, or RF) lead to a decrease in the mechanical and thermal stability of IVD tissues, which, in a number of cases, is the cause of postoperative complications when exposed to intense energy sources with excessively increased or poorly controlled power. Therefore, the main target of laser action in the LRD technology was chosen to be the NP of the disc and the EP region without a significant effect on the AF [32].

### 2.2. Goals, Objectives, and Targets of Laser Treatment

Targets for laser effects and possible types of cartilage reactions to laser radiation were discussed in detail in [34,35,42] and are shown in Figure 2.

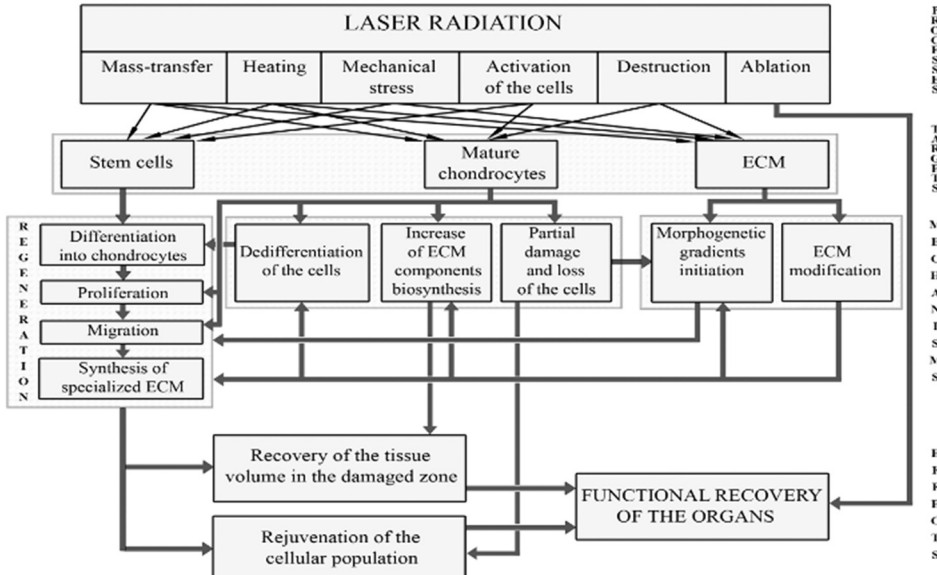

**Figure 2.** Targets for laser radiation and regeneration processes leading to the restoration of IVD.

Laser radiation can directly affect (a) cells; (b) various components of the ECM; (c) signaling molecules produced by cells; (d) intercellular interactions; (e) differentiation and dedifferentiation of cells, their migration and biosynthetic activity. Possible processes and pathways for cartilage regeneration include: (1) additional supply of cells; (2) enhancement of biosynthesis of ECM components; (3) stimulation of mature chondrocytes; and (4) activation of resident stem cells in the direction of their proliferation, differentiation, and production of ECM. Unlike ablative or low-energy laser treatment, modifying laser radiation causes controlled thermal and mechanical effects (both on cells and on ECM), which leads to activation of tissue regeneration. Spatio-temporal modulation of laser radiation allows to control the actual distribution of stretched and compressed zones in the cartilage.

Mechanical loads are important factors governing the chondrogenesis orchestra, including the processes of cell differentiation [43]. Thermomechanical laser action can play a decisive role in the differentiation of immature cartilage cells. The advantage of laser action on the proliferation of chondrocytes in comparison with other thermal, mechanical, and chemical effects was demonstrated in [44]. Laser radiation can stimulate the processes of proliferation and the acquisition of a specialized phenotype by resident or mesenchymal stem cells, promoting their transformation into mature hyaline-like chondrocytes. It is important that laser modification of the fine structure of the ECM does not change its overall organization. This provides a natural environment for chondrocytes and leads to the restoration of hyaline-type cartilage [32,42].

### 2.3. Mechanisms of Laser Regeneration

Laser activation of cell regeneration can occur as a result of the direct action of laser radiation on cells, and indirectly—through the ECM due to the modification of its structure and the formation of temperature and mechanical stress fields. It is known that chondrocytes are sensitive to environmental conditions, in particular, to temperature and mechanical stress [45,46]. Modulated in space and time, laser radiation causes pulsed-periodic heating, leading to inhomogeneous thermal expansion and inhomogeneous pulsating field of mechanical stresses, which can actively affect the function of chondrocytes, contributing to their proliferation and biosynthetic activity. Compressive stress of a certain frequency (0.1–10 Hz) and amplitude (5–25 MPa) promotes the proliferation of chondrocytes and the activation of the synthetic activity of cells to produce collagen and proteoglycans [47,48]. Going beyond the indicated boundaries of the vibration frequency range and an excessive increase in the amplitude of mechanical action leads to inhibition of the life of cells and to their apoptosis [48].

Mechanical forces act on processes in the cells through mechanotransducers PIEZO1 and PIEZO2, which convert external mechanical stresses into electrical signals that control cell metabolism and apoptosis [49,50]. Another important mechanism of laser regeneration is the formation and distribution of thermo-shock proteins, cytokines, and growth factors, released by a small fraction of cells dying as a result of laser action [51]. According to the data [52,53] obtained in experiments with cultures of chondrocytes, growth factors and cytokines increase the production of proteoglycans and type II collagen, cause accelerated cell proliferation, and inhibit their apoptotic death. In experiments [54], the induction of chondrogenic differentiation of bone marrow stem cells under the influence of growth factors and glucocorticoids were found. In vivo studies, an increase in IVD mass and proteoglycan content and the appearance of cell clusters similar to clusters of normal hyaline cartilage were found [55].

The third important mechanism of laser regeneration is the modification of the porous structure of the ECM (Figure 3). Loosening of the matrix is observed using an optical or ultrasound microscope [40,56], and a more precise microscopy (atomic force microscope, structured irradiation microscope, and electron microscope) makes it possible to study specific pores of micron and submicron size [57]. Micropores increase the diffusion of water and nutrients in the cartilage matrix and thus promote regeneration. In this case, there are no macroscopic structural changes and deterioration of the mechanical properties of the cartilaginous tissue [32,40]. The pore size in ECM is of great importance. Macroscopic pores and defects (ranging in size from hundreds of microns to several millimeters) tend to become overgrown with cells and matrix. Small pores a few microns in size, as a rule, do not overgrow. Such micropores, formed in the cartilaginous tissues of the IVD and joints of animals as a result of laser irradiation, were observed several months after laser exposure [42,57].

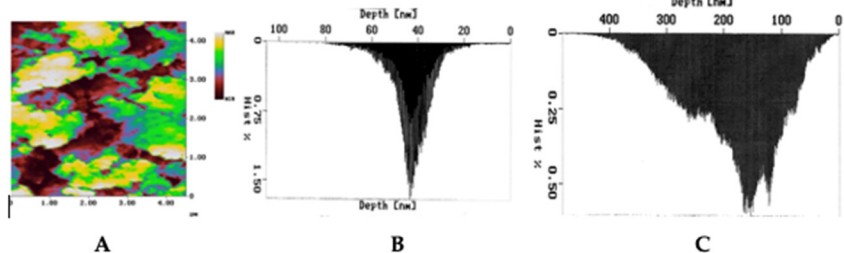

**Figure 3.** Formation of micropores in the rabbit cartilage plate under laser action studied by Atomic Force Microscopy: (**A**) image of micropores; size distribution of micropores (**B**) before and (**C**) after laser exposure.

The regeneration processes discussed above are associated with a slight heating of the tissue and can be called thermomechanical mechanisms. Regarding the mechanisms of photobiomodulation, low-level laser therapy (LLLT) does not lead to noticeable heating of the tissue. Various hypotheses have been put forward concerning various chromophores [58–62]. The effects of laser radiation over living tissues are based on the absorption of its energy and its transduction into a biological process, mainly due to Adenosine triphosphate (ATP) synthesis. LLLT may modulate transcription of several growth factors, including fibroblast growth factors (FGF) and vascular endothelial growth factors (VEGF) [63]. There is evidence that both ATP activation and cell proliferation under the influence of R-NIR light occur through the interaction of photons with intracellular water layers. The interaction has at least two biologically important effects: a change in density (volumetric expansion) and a decrease in the viscosity of water [64]. LLLT can increase proliferation of the marrow stem cells without producing high levels of reactive oxygen species [65]. Bone marrow irradiated in vivo can be used to treat a variety of disease conditions. The irradiation may be performed by transcutaneous application of infrared laser radiation over the area of a marrow-containing bone. It was found that labeled cells were seen in and near the infarct up to eight weeks post myocardial infarction, while none

were seen in sham-operated hearts. It was concluded that following myocardial infarction, mesenchymal stem cells are signaled and recruited to the injured heart, where they undergo differentiation and may participate in the remodeling process [66].

A detailed analysis of photomodulation processes is beyond the scope of this work. Here, we just note that a comparative study carried out in [47] showed that the thermomechanical effect of the laser radiation causes a more pronounced stimulation of chondrocytes than photomodulation. A recent review of publications on the use of lasers in the treatment of pain syndrome of the musculoskeletal system also showed certain advantages of moderate radiation power compared to low-intensity laser exposure [67]. The conclusion made in [68] seems to be important that "Treatment of pain syndrome of the IVD is possible without complete recovery of age-related and degenerative changes in the NP by accelerating the healing of damage to the periphery of the disc by stimulating cells, accelerating the transport of metabolites and preventing adhesions and repeated injuries". This approach corresponds to the LRD technology, which not only reduces pain, but leads to the restoration of cartilage of IVD with partial replacement of the NP tissue with hyaline cartilage. Laser radiation modulated in space and time allows for precise control of various parameters important for the process of structure modification and regeneration, i.e., temperature, amplitude, and frequency of mechanical action, as well as mass transfer into cells, which leads to the appearance of morphogenetic gradients and control of tissue regeneration.

### 3. Laser Reconstruction of the Intervertebral Disc in Animal Experiments

*3.1. Morphological Changes in Rabbit Discs as a Result of Ex Vivo and In Vivo Laser Irradiation*

The effect of laser radiation on the processes of IVD tissues regeneration was studied based on theoretical models of temperature fields and thermomechanical stresses arising during laser action on cartilage [69,70] and experimentally ex vivo and in vivo on rabbits using holmium (2090 nm) and erbium fiber (1560 nm) lasers. To simulate the DDD, the discs were preliminarily mechanically damaged by removing the NP or puncturing the AF with a needle. A month after the mechanical damage of the discs, they were irradiated using an optical fiber of 400 μm in diameter, which was introduced into the IVD through a puncture needle. The irradiated area was examined by optical coherence tomography and histology methods one and two months after the operation. It was shown that, depending on the laser settings, the damaged tissues are transformed into various types of cartilage (hyaline or fibrous) or bone tissue is formed [30,31].

Non-ablative laser action modifies the cartilage structure, promotes the growth of the new tissue with a pronounced heterogeneous structure, which causes a useful functional combination of elastic and plastic properties of IVD [32]. It has been shown that radiation with a wavelength of 1560 nm causes changes in areas larger in size (about 1 mm) than radiation of 2090 nm. Temporal and spatial dependences of temperature and thermomechanical stresses in the affected zone were studied in ex vivo experiments [47,66]. It is shown that when exposed to laser pulses 1 s in duration, with a frequency of 0.5–1 Hz at a radiation power of 1 W, stresses (pressures) from 10 to 25 MPa arise. It is known that mechanical stress is one of the main stimuli for the activation of cartilage cells [43]. Usually, physiological loads that increase the synthetic activity of chondrocytes do not exceed 20–25 MPa. At 30–50 MPa, the formation of shock proteins was observed, and even higher pressures can cause noticeable damage and cell death [46,71].

Then, in a short (3-day) in vivo experiment, a preliminary assessment of 12 different modes of laser exposure was carried out. It was found that the repetitively pulsed laser radiation with frequencies in the range of 0.3–2 Hz leads to more pronounced regeneration processes than continuous wave radiation. The intensity of cell proliferation and synthesis of ECM components after laser exposure was significantly higher than after mechanical destruction of the inner region of the disc. Three modes of laser radiation (given in Table 1) with a pulse duration of 10, 100, and 1000 ms and a repetition rate of 0.5 and 1 Hz, for which the maximum arising stresses were in the range 15–25 MPa, and the maximum tissue heating was 20–30 °C within 10 s did not lead to a noticeable denaturation of IVD

collagen. These laser settings were selected for further research. At this stage, the DDD of the rabbit was simulated by puncturing with an 18 G needle. A month later, this led to the development of a degenerative process with the formation of zones of necrosis, which remained in the IVD for a long (at least 6 months) postoperative period. Then, laser treatment was performed to study the most pronounced biological effects leading to cartilage regeneration and IVD restoration. (Figure 4A).

**Table 1.** Laser mode selection for clinic. Wavelength 1560 nm, number of pulses in a series 10.

| Laser Mode | 1 | 2 | 3 |
|---|---|---|---|
| Pulse duration, ms | 1000 | 100 | 10 |
| Pulse repetition rate, Hz | 0.5 | 1 | 1 |
| Power, W | 0.7 | 1.5 | 3 |
| Number of the series of pulses | 3 | 3 | 1 |
| The duration of the pause between the series of pulses, s | 15 | 10 | - |

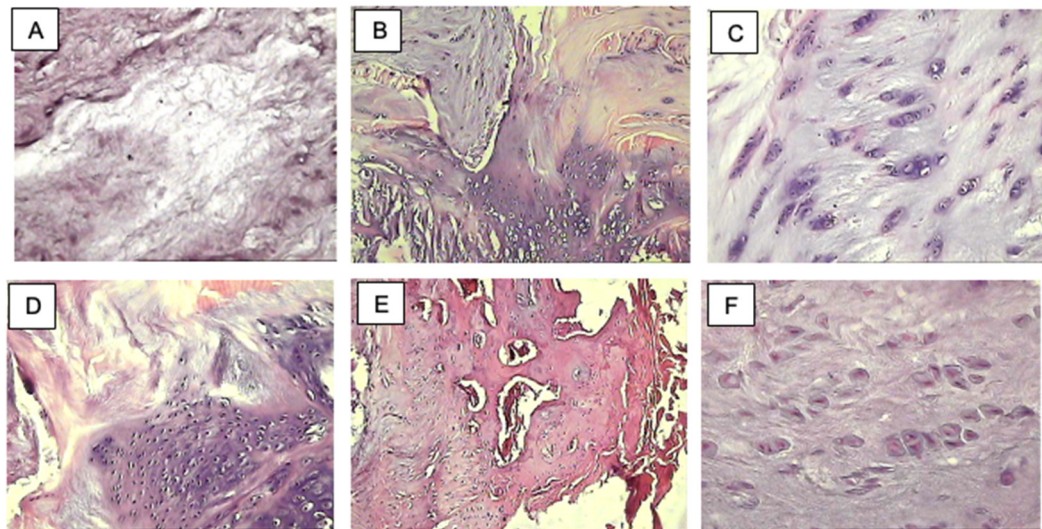

**Figure 4.** Histological examination of the in vivo laser restoration of damaged discs in rabbits: (**A**) simulation of DDD (control): NP in 2 months after needle insertion. Necrotic acellular tissue, ×400; (**B**) laser-irradiated disc, mode 1. In 30 days in the upper left corner is a fragment of fibrohyaline cartilage in the NP. Area of necrosis (top right) and growing hyaline cartilage tissue (bottom), ×100; (**C**) irradiated disc, mode 1. In 90 days. Transitional fibrohyaline cartilage in the NP. Numerous isogenic groups, fibrosis and hyaline-like (below) matrix, ×200; (**D**) irradiated disc, mode 2. 30 days. Development of hyaline cartilage in two months after laser irradiation, ×400; (**E**) irradiated disc, mode 2. 30 days. Osteophyte, consisting of bone tissue (right), fibro-hyaline cartilage (left), ×200; (**F**) irradiated disc, mode 2. 90 days. Fibrous-hyaline cartilage in the NP, ×400. Hematoxylin and Eosin (H&E).

*Results of laser exposure in the selected modes.*

Mode 1: On the 30th day, hyaline EP thickness increased significantly. At the same time, rounded cells with lacunae (mature chondrocytes) prevailed in the outer layer while the cells without lacunae (i.e., proliferating chondroblasts replacing NP cells) occupied the inner layer. In the middle and internal zones of AF, there was a pronounced fibrohyaline transformation of necrotic tissues. In some animals, the normal tissue of the NP with a reticular ECM and clusters of notochordal cells was partially retained. In most cases, the NP tissue was partially or completely replaced by fibrohyaline or hyaline cartilage (Figure 4B). In the 90 days after the operation, fibrohyaline and hyaline cartilages predominated in the NP of all animals (Figure 4C). Signs of necrosis in this mode were minor.

Mode 2: After 30 days, the formation of hyaline or fibro-hyaline cartilage (Figure 4D) was observed in the NP, while, in the outer third of the AF, osteophytes formed consisting of cancellous bone tissue and fibro-hyaline cartilage (Figure 4E). The hyaline layer between the bone part of the EP and IVD was thickened. In the middle and inner layers of the AF, the structure slightly changed in some of the discs. After 90 days, the inner layer of AF was largely replaced by fibro-hyaline or hyaline cartilage, and in the middle layer, manifestations of tissue remodeling were observed, associated with the gradual formation of fibrohyaline cartilage. Signs of gradual tissue transformation into fibrohyaline and hyaline cartilage were revealed in the NP. After 30 days, diffuse compaction of the NP matrix occurred, the size of cell clusters decreased, a change in the dominant cell type was noted—instead of notochordal cells, chondrocyte-like cells began to prevail in the NP. A total of 90 days after surgery, most of the NP (up to 90% of the volume) had already been replaced by fibrohyaline cartilage, in which signs of hyaline cartilage were more pronounced in some areas (Figure 4F), and signs of fibrous cartilage were in others. The border between the NP and AF was hardly identifiable. In AF, foci of hyaline or fibrohyaline cartilage with pronounced proliferation of chondrocytes and the formation of large isogenic groups were observed.

Mode 3: In the 30 days after irradiation, large fields of fibrohyaline and hyaline cartilage with isogenic groups were formed in the middle and inner layers of the AF, as well as at the site of the NP. In several cases, foci formed by the tissue of the NP appeared in the AF, possibly due to the "throwing" of the NP particles under irradiation with a short laser pulse. After 90 days, there was a pronounced expansion of the EP and a widespread transformation of tissue into fibro-hyaline and hyaline cartilage, which had a more diffuse character than at 30 days.

Thus, the histological and histochemical study of IVD under all modes of laser irradiation showed that after 30 and 90 days, similar morphological changes occurred, which lead to the replacement regeneration of tissue damage in the NP and AF. In the NP, the formed areas of necrosis gradually decreased in size due to the outgrowth of proliferating chondroblasts, which gradually matured into chondrocytes, the sources of which were both surviving chondrocyte-like cells and undifferentiated cells of the hyaline EP. As a result, areas of necrosis are replaced by tissue that is nonspecific for the NP: fibrohyaline cartilage, which is a transitional form from more fibrous to more hyaline, and hyaline cartilage, which is associated with the nature of the source cells. In AF, the most significant necrotic and dystrophic changes occurred in the inner and transitional layers at the early stages. To a lesser extent, they are present in the middle layer of the AF. Changes began with the proliferation of active hypertrophic chondrocytes with many isogenic groups that replaced the affected tissues with fibrous cartilage with a non-lamellar structure or, which happened more often, with fibrohyaline cartilage. The source of the cartilage formation was probably undifferentiated AF chondrocytes. Hyaline cartilage developed at the AF/NP border or near the hyaline layer of the EP.

The laser irradiation dosimetry affected the rate of transformation of the disc tissue, as well as the volumetric and topical ratio of areas with different histological patterns. Since the above-described processes in NP and AF under different modes of laser irradiation occur at different rates, but in one direction (genetically predetermined by the staging of differentiation of cells of the chondrocytic series), than the difference between the biological effects of different modes was somewhat decreases three months after exposure.

An important feature of laser-induced regeneration of cartilaginous tissue was that laser irradiation caused tissue transformation not only in areas directly in the irradiated zone, but also in zones located at a significant distance (several mm) from the affected zone. Under laser irradiation, the high activity of regeneration processes was apparently due to a regulatory thermomechanical response of poorly differentiated cells. The laser radiation activated the cells of the "resource" areas of the disc—the sources of the young cellular elements of the chondrocyte type—the inner and transition layers of the AF, the hyaline tissue of the EP and part of the cells of the NP. Young cells with a high proliferative and

biosynthetic potential migrated from the listed zones to areas located near the damaged zones and provided acceleration and a qualitative advantage for the regeneration observed during pulsed-periodic laser exposure.

There are two factors in DDD treatment: pain alleviation and the actual healing of the IVD. It is proposed that LLLT can regulate the inflammatory reactions, modulate the secondary damage, and reduce programmed cell death and edema in the primary phase of recovery [72]. High energy densities leading to damage of neuroreceptors also can lead to pain relief, but this effect is not long lasting and does not result in IVD restoration [32]. The thermomechanical effect of moderate power density laser radiation can positively influence the behavior of cells and neuroreceptors via PIEZO proteins, but this hypothesis requires future focused research. In addition, there is reason to believe that the reduction in disc instability after LRD gradually leads to a decrease in pain. Most animals do not walk upright, the load on the IVD is not so great, and the process of inactivation of IVD pain receptors in an in vivo animal experiment is difficult to objectify. We studied only the actual healing of IVD in rabbits. The combination of macroscopic and histological studies allowed us to conclude that Modes 1 and 2 are acceptable for the clinic, in which the NP and the inner layer of AF are most transformed into hyaline and fibrohyaline cartilage, which stabilizes the spine under conditions of degenerative destroyed IVD. Mode 1 with a pulse duration of 1 s, a frequency of 0.5 Hz, at which the maximum heating of the tissues of the NP was 20–25 °C, and the arising pressures were 20–30 MPa, and which gave the most stable results of IVD restoration without the formation of osteophytes and noticeable denaturation of tissues, was selected for clinical research.

### 3.2. Safe and Effective Laser Settings. Choice of Optimal Parameters of Laser Radiation

The transition from optimal irradiation regimes causing IVD restoration in rabbits to the regimes of irradiation of discs of large animals and patients required special studies. It was assumed that the main mechanism of the therapeutic effect is associated with a small short-term and local heating of the disc and the resulting thermomechanical stresses, which are determined by the values of the achieved temperature gradients. Three modes of laser exposure, leading to heating of rabbit discs by (1) 30–35 °C (radiation power 1.2 W); (2) 20–25 °C (0.9 W); (3) 7–10 °C (0.6 W) were tested ex vivo on sheep discs. The radiation of a fiber laser with a wavelength of 1560 nm was applied through a fiber of 0.6 mm in diameter. The pulse duration was 2 s, the interval between the pulses was 1 s, the irradiation was carried out with three series of pulses, 10 pulses in each series, and the interval between the series was 20 s.

Histological and histochemical examinations of the irradiated discs showed that in the given time regime of irradiation at a power of 0.6 W, the structure of the disc differed little from the initial structure of the unirradiated NP. With an increase in power to 1.2 W, a noticeable denaturation of the disc tissues was observed, the area of coagulation necrosis increased to 10–15%. With a power of 0.9 W, changes in temperature by 20–25 °C correspond to the final temperature of the disk 55–60 °C, at which the proportion of coagulation necrosis is insignificant, a very small fraction of cells are damaged, which nevertheless emit signaling molecules that trigger regeneration processes into the surrounding matrix. It is this regime of laser exposure in experiments on rabbits in vivo that led to the predominant growth of hyaline-type cartilage. Therefore, the temperature range of 20–25 °C was determined as the optimal heating for laser treatment of discs, since at the selected time irradiation mode, such heating does not lead to significant coagulation of the tissue and is sufficient to create the necessary thermal stresses leading to the activation of chondrocytes, to loosening of the ECM, formation of micropores, generation of shock proteins, growth factors and other signaling molecules that promote cartilage regeneration.

To achieve optimal heating (by 20–25 °C) of human discs, the modes of laser exposure were used, which were previously studied in the animal experiments. The radiation power was refined considering the differences in the sizes and optical properties of human and animal discs. Since, with a decrease in the water content, the heat capacity of collagen

decreases, which compensates for the change in light absorption and leads to a weak dependence of the optimal radiation power on the water content. A more important parameter affecting temperature was found to be the size of the discs, which are significantly different for humans and most of the animals [73]. For relatively large human discs, the radiation energy spreads by thermal conduction to a larger volume than for the discs of rabbits and sheep; therefore, the optimal power values for human discs should be higher than for these animals. Calculation based on the solution of the thermal conductivity equation shows that in the selected irradiation mode, to heat human discs with a diameter of 4–5 cm by 20 °C, requires 25–30% more power than for the same heating of a sheep disc with a diameter of 1.5–2 cm. Therefore, for patients undergoing LRD operations for the back discs, the average optimal power was chosen to be 1.2 W (30% more than for a sheep), and for neck discs with a diameter of 2 cm, which are close in size to the discs of a sheep, the optimal radiation power was selected 1 W. Thus, experimental studies on animal discs established an adequate model of DDD, demonstrated the possibility of laser disc repair with the formation of hyaline and fibrohyaline cartilage, which ensures the stability of the spine. Calculation and measurement of the temperature dynamics in the irradiation zone, as well as histological and histochemical studies, made it possible to determine and clarify for human discs the optimal modes of laser exposure for human IVD, in which the disc is repaired without significant tissue coagulation in the irradiated zone.

## 4. Clinical Aspects of LRD of IVD in DDD of the Spine

### 4.1. LRD Puncture

LRD is a new approach to the treatment of DDD, based on the control of mechanical stress in tissues that have been exposed to local, short-term non-destructive laser irradiation. After detailed studies on animals and clinical studies on a selected group of patients, the LRD medical technology received the approval of the Ministry of Health of the Russian Federation and has been regularly used in the clinic since 2006. Currently, LRD is one of the most common effective methods for the prevention of degenerative diseases of the spine. LRD was applied to more than 5000 patients with 92.5% positive effect over 5 years of follow-up.

*Indications for puncture LRD* are: 1. Chronic discogenic pain syndrome, which cannot be stopped by conservative therapy; 2. Degenerative changes in IVD grade 2–4 according to the Pfirrmann MRI classification [74]. A more accurate diagnostic procedure is provocative discography or its version of CT discography. Thus, the detection of concordant pain syndrome using discography in combination with DDD according to MRI data is an indication for LRD.

*Contraindications for puncture LRD* are: 1. Compression of nerve structures with radicular pain syndrome or spondylogenic myelopathy; 2. Significant decrease in the height or fibrosis of the IVD. 4th degree according to the Pfirrmann classification [74]; 3. Absence of signs of DDD on MRI and discography (first degree according to the Pfirrmann classification); 4. Local or generalized inflammatory process; 5. Disorders of the blood coagulation system and untreated coagulopathy; 6. Significant psycho-emotional component in pain syndrome, including discography intolerance.

*Technique of LRD* [32–34]. The scheme of LRD is presented in Figure 5. The puncture LRD using optimal laser dosimetry is performed under local anesthesia using continuous X-ray control in full contact with the patient. This makes it possible to completely avoid cases of trauma to the nerve structures and to evaluate the effect of exposure to the disc during the irradiation itself. To prevent the development of complications, antibiotic prophylaxis is performed (1 g of Ceftriaxone 30 min before the operation). Before the intervention, a provocative discography is performed to identify the degree of concordance of pain, as well as the severity of degenerative lesions of the IVD. The water-soluble contrast Omnipak-300 is used, which provides good visualization of the IVD during discography. Laser irradiation of the disc is carried out using a 1560 nm fiber laser with feedback control system (Arcuo Medical Inc., Los Altos, CA, USA) [75]. Optical fiber with a diameter of 600 μm is inserted

into the IVD through a puncture needle G18, and irradiation of three or four zones of the NP is performed in the following mode: pulse duration 2 s, interval between pulses 1 s, and the number of pulses in series is 10; interval between series 20 s, the number of series is three. Laser power is 1.2 W and 1.0 W for the back and neck discs, respectively.

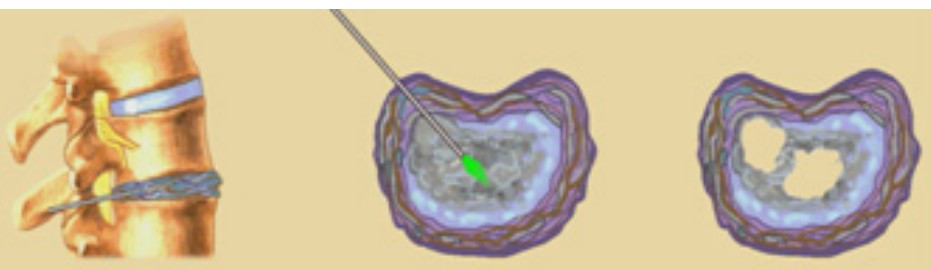

**Figure 5.** Scheme of LRD technology.

*Cervical spine*. Puncture of the IVD of the cervical spine is carried out according to the method of Cloward [76]. The patient is placed in a supine position with a bolster under the shoulders and the head thrown back (Figure 6). The needle insertion point is in the groove between the trachea and the sternocleidomastoid muscle The side of the puncture depends on the presence of previous surgical interventions, deforming scars and the preference of the surgeon. In the cervical spine, one central zone of the disc is exposed to radiation, and two at the sites of the transition from the NP to the AF.

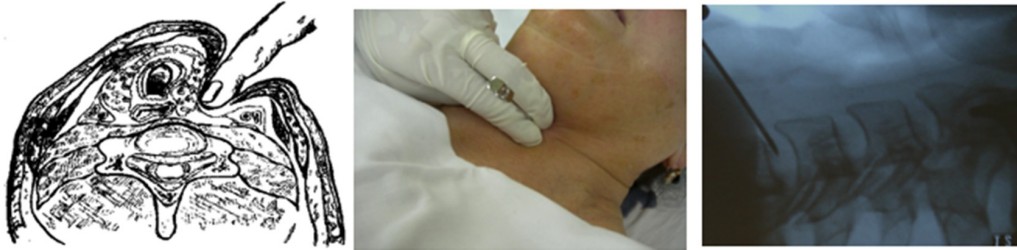

**Figure 6.** LRD of the cervical spine according to the Cloward approach.

*Lumbar spine*. Puncture of the lumbar intervertebral discs is performed with the patient lying on his stomach. The needle is inserted into the lumbar discs from the postero-lateral approach. The needle is inserted up to the AF, the course of its installation is controlled by an image intensifier in the fluoroscopy mode (Figure 7). After piercing the AF, the needle is inserted another 2 cm and placed in the center of the disc. After the discography, subject to the necessary indications, an optical fiber is inserted into the disc for conducting laser radiation. In total, three or four zones of the IVD are irradiated in the direction of the needle placement, one or two zones are in the center of the NP and two zones along the inner surface of the AF.

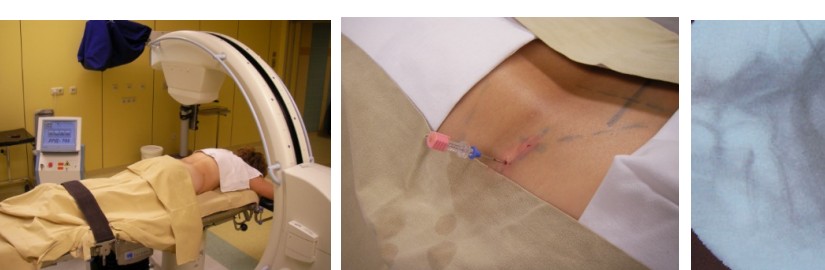

**Figure 7.** LRD of the lumbar spine.

*Clinical effect of puncture LRD*. Immediately at the time of laser irradiation of the IVD, in some cases, patients note light sensations of "tingling, warmth, bursting" in the spine. In the days following puncture LRD, the majority of patients experienced relief of existing pain. This is most likely due to local heating and microcavitation effect on the pain receptors of the disc. The exacerbation of pain syndrome observed in some cases was associated with a needle puncture of the inflamed AF. Usually, within a few weeks, the aggravation of pain in these patients diminished, and the level of pain becomes significantly less than preoperative. Complications in the postoperative period were not observed if all the rules were followed. Persistent positive effects of LRD are associated with the regeneration of the intervertebral disc cartilage and begin to appear approximately 3–6 months after the intervention. They consist of a decrease in pain, an increase in the voluntary range of motion in the spine [32,34].

Long-term results of LRD have been analyzed in several long-term studies [33–37]. According to the results of one of the studies (475 patients), the total score before the operation was 33.0 (Q25% = 23.0; Q75% = 42.5), after the operation it significantly increased (Z, $p < 0.001$), reaching 64.0 (Q25% = 50.0; Q75% = 78.5). The median index of pain severity according to the visual analogue scale (VAS) before surgery was 7 points (Q25% = 5.0; Q75% = 8.0), and after LAD it decreased to 3.0 points (Q25% = 2, 0; Q75% = 4.0). For the purpose of the general assessment of the efficiency of LRD, the following gradations of results were used: "Positive result"—the total score SF-36 after the LRD is higher than before the operation, the VAS indicator after the LRD is lower than before the operation; "Negative result"—the total score SF-36 after the operation is lower than before the operation, the VAS after the operation is higher than before the operation; "Neutral result"—the total score SF-36 after the LRD did not change compared to the preoperative level; VAS after surgery did not change compared to the preoperative level. When performing LRD on the discs of the cervical spine, the results were distributed as follows: positive in 91.6% of cases; negative and neutral—4.2% of cases. When performing LRD on the discs of the lumbar spine, the following results were obtained: positive 89.1% of cases; negative 4.1% of cases; neutral 6.8% of cases.

*Example 1.* Male patient, 53 years old, suffering from chronic low back pain for many years (VAS 7/10). MRI revealed degenerated L4L5 disc, Pfirrmann 5 grade. The patient underwent provocative L4L5 discography which detected highly concordant low back pain and succeeded with punction transforaminal LRD of L4L5 disc. Pain intensity decreased considerably in 24 months (VAS 3/10), and he could continue working activity. Control MRI revealed drastic improvement in MRI disc morphology as change of the disc degeneration to Pfirrmann 4, apparent formation of the new cartilage tissue in the center of the disc, increase in signal intensity from the disc cartilage on the T2 WI and STIR images (Figure 8). Additionally, the change of the signal is noticeable, from the adjacent to the disc vertebral bone as the mark of disc metabolism shift in the T1 WI image.

In addition, the data of histological and electron microscopic examination of biopsy samples of IVD tissues were obtained from three patients in two or three years after LRD (Figure 9). In the irradiated part of the discs, pronounced regenerative processes were found in the form of proliferation of chondrocytes and neoplasms of cartilage tissue of the fibrous-hyaline type. In the non-irradiated part of the discs, the replacement of the disc tissue with degeneratively altered nonspecific fibrous cicatricial connective tissue continued [33].

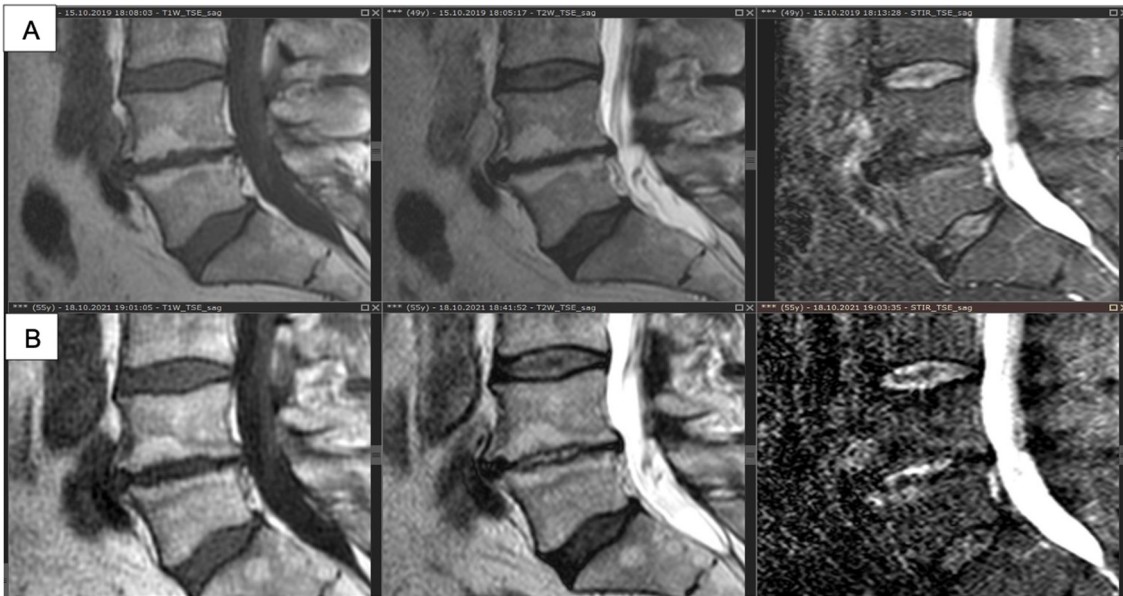

**Figure 8.** MRI results for a patient 53-year-old in 24 months after LRD: (**A**) before and (**B**) after punction transforaminal LRD of L4L5 disc. T1WI—**left column**, T2 WI—**center column**, STIR—**right column**. Change of the disc degeneration from Pfirrmann 5 to Pfirrmann 4 (T2 WI), formation of the new cartilage tissue in the center of the disc (T2 WI, STIR), change of the signal from the adjacent to the disc vertebral bone as the mark of disc metabolism shift (T1 WI).

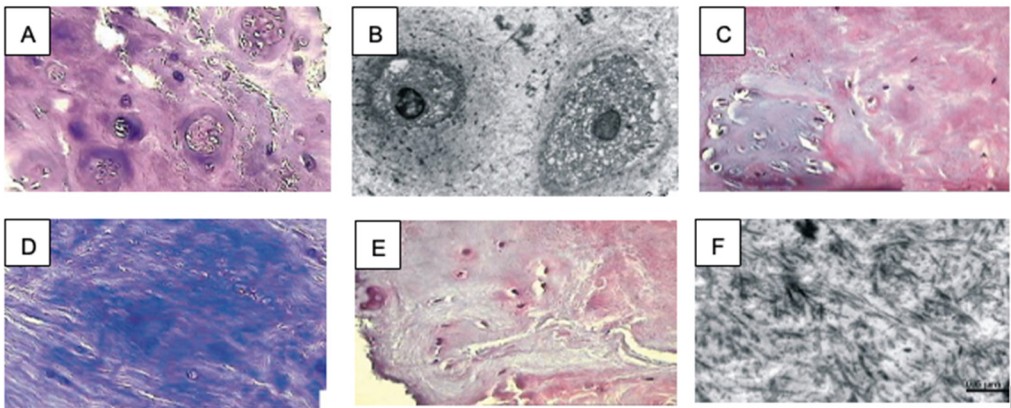

**Figure 9.** Morphological structure of the IVD of a patient in 2.5 years after LRD: (**A**) numerous isogenic groups of chondrocytes with high levels of PGs. Toluidine Blue, ×400; (**B**) biosynthetically active chondrocytes in isogenic groups, thin CFs and PGs around the cells in the ECM. SEM, ×6000; (**C**) on the left is the transition area of fibrous cartilage into hyaline ones. H&E, ×400; (**D**) in the center-bright metachromatic areas. Groups of chondrocytes of the type of hyaline cartilage cells are visible. Toluidine Blue, ×400; (**E**) on the left is a fibro-hyaline cartilage: chondrocytes in lacunae and homogeneous, in places fine-fibrous, basophilic ECM. H&E, X400; (**F**) CFs and PGs are randomly located in the ECM of hyaline-like cartilage. SEM ×40,000.

*4.2. LRD after Surgical Interventions on the Spine as a Means of Preventing the Development of Recurrent Disc Herniation, Instability of the Affected Segment and Further Development of the DDD*

Once started, DDD progresses over time. When observing the natural course of the degenerative process after 2 years, the deterioration of the disc condition by 1–2 points according to Pfirrmann was observed in 47.6% of patients, and in the next 5 years, it reached 95.2% [77]. At the same time, the actual intervention on the IVD to remove symptomatic hernial protrusion can accelerate the degenerative process. During long-term follow-up from 5 to 25 years for patients after open classical discectomy, based on radiological changes,

acceleration of disc degeneration was noted in 48.7% of cases, microsurgical technique reduces this risk, but not completely, provoking such negative changes in 9.1% [15]. This leads to the progression of pain syndrome, and in 7–15% of patients it leads to recurrence of herniated discs. Removal of an intervertebral hernia by any method, including puncture, endoscopic or microsurgical, leads to the inevitable progression of DDD due to iatrogenic mechanical action on IVD structures. The minimal puncture of a healthy IVD in a rabbit causes delamination of its AF and, after 12 weeks, visible degenerative changes in the disc on radiographs [78]. Puncture of an IVD with a needle serves as a standard model of its degenerative lesion in experimental studies [79].

LRD induces a regenerative response of the cartilaginous tissue, accompanied by the formation of new hyaline and fibrohyaline cartilage in the disc cavity. The possibility of combining laser irradiation of the disc with the procedure for removing a disc hernia makes the laser technique a valuable addition to the surgical intervention and transforms it from ablative to regenerative, when the disc tissue is not only removed, but also internal trophic processes in the IVD are stimulated, leading to disc regeneration [34]. In this case, the LRD is performed through the operating wound after the completion of microsurgical removal of the hernial protrusion and decompression of the nerve structures. To do this, an 18G needle is inserted into the disc through the natural hernial orifice in the defect of the AF. Before needle insertion, medial traction of the root and dural sac is performed to expose the disc. If the defect in the AF is closed by an intact posterior longitudinal ligament, then the needle is inserted by piercing it. After inserting the needle and removing the mandrel, a laser fiber is inserted into it. Then, the disc is irradiated according to the standard protocol in three zones of the NP: in the center, at its proximal and distal border with the AF along the disc puncture line.

*Example 2*. Patient, 44 years old, suffered from acute S1 radiculopathy against the background of hernial protrusion of the L5-S1 disc of the vertebrae. On MRI of the patient before the operation, there is a pronounced degenerative lesion of the L5S1 disc of grade 4 according to Pfirrmann [74]. During the operation, after removal of the hernia, intraoperative laser reconstruction of the L5S1 IVD in three areas was performed according to the standard protocol. In the postoperative period, the patient showed excellent results. During the follow-up, the MRI showed not only a full decompression of nerve structures, but also a visible regenerative IVD response, during which the MRI showed positive dynamics with a decrease in the gradation of degenerative lesion of the L5S1 disc up to the third after 6.5 months and up to 2nd after 3 years, when a well-formed homogeneous central nucleus was formed in the IVD adjacent to the irradiated disc. This zone of contact between bone and cartilage tissue is important for metabolic processes in the disc and reflects it. In Figure 10, T1 VI shows an increase in the signal from the subcortical layer of the vertebral body near the disc. This means an increase in fluid metabolism and may indirectly indicate an increase in disc metabolism.

Currently, endoscopic discectomy is becoming the new standard in the treatment of IVD hernias. The technique of percutaneous endoscopic discectomy using a single-portal approach and an endoscope no more than 7 mm in diameter takes full advantage of the benefits of endoscopic surgery. In this case, the iatrogenic postoperative cicatricial process is minimal both in the epidural space and in the paravertebral muscles and tissues. This allows for achieving excellent clinical results, reducing the time spent in the hospital to one day, and restores the patient's ability to work as early as possible. Modern techniques of percutaneous lumbar discectomy are implemented through transforaminal and intercutaneous access. The development of these techniques is associated with the names of Hijikat, Kambin, and Ruetten [80–82]. After the endoscopic removal of the sequestration is completed, the final stage of the LRD operation on the disc is performed. In the case of an interstitial approach, the introduction of a laser guidewire is performed without a needle under visual endoscopic control into the area of the AF defect. In transforaminal surgery, laser irradiation can be performed similarly through an endoscope, or through a separate

puncture with a transforaminal needle after removal of the endoscopic port. In both cases, the irradiation protocol is identical to that described above.

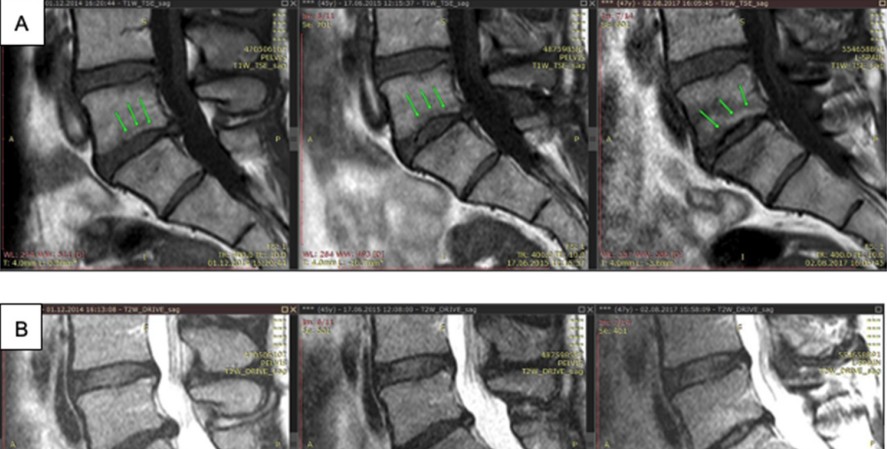

**Figure 10.** Combination of LRD and microscopic discectomy for a female, 44 y.o., with L5S1 disc herniation: (**A**) MRI T1 WI. Preoperative (**left picture**), 6.5 months after surgery (**center picture**), 3 years after surgery (**right picture**). Change of the signal intensity at the adjacent to the disc bone (arrows) manifests an increase in disc metabolism; (**B**) MRI T2 WI. Disc degeneration after Pfirrmann [74]: Grade 4—preoperative (**left picture**), Grade 3–6.5 months after surgery (**center picture**), Grade 2—in 3 years after surgery (**right picture**).

*Example 3.* Combination of LRD with percutaneous lumbar transforaminal endoscopic discectomy. A 51-year-old patient with hernial protrusion at the level of L4L5 vertebrae with clinical manifestations of discradical conflict and DDD of grade 4 according to Pfirrmann [74]. Percutaneous lumbar transforaminal endoscopic discectomy with LRD was performed at the level of L4L5 vertebrae from the right-sided approach. Immediately after the LRD, a persistent regression of pain syndrome was observed, and after 9 months, in addition to the clinical effect, the control MRI showed a pronounced positive dynamic of the degree of DDD according to Pfirrmann—from 4 to 2 degrees. At the same time, the T2 VI MRI mode showed a pronounced development of cartilage tissue in the disc cavity (Figure 11).

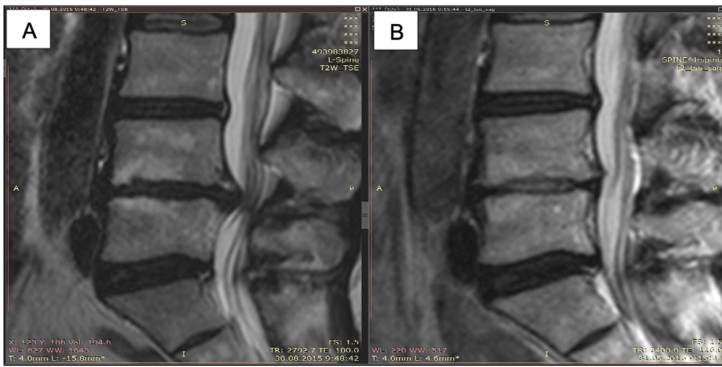

**Figure 11.** Combination of LRD and transforaminal endoscopic discectomy for a female, 51 y.o., L4L5 disc herniation, MRI T2 WI: (**A**) Preoperative—disc degeneration: Pfirrmann Grade 4; (**B**) postoperative—disc degeneration Pfirrmann Grade 2. DDD decreased and formation of the new cartilage tissue is seen.

Thus, the combination of LRD with discectomy performed in a different way (microsurgical, endoscopic, or puncture) leads to regenerative changes in the disc. This is manifested by an increase in the intensity of the signal from the disc on MRI with a decrease in the degree of its degenerative damage. This combined surgery is an effective way to prevent the progression of DDD.

Currently, most of the beneficial effects have been obtained in the study of the restoration of cartilage tissue of the spine and joints. The preliminary results of the research prove the possibility of regeneration of other tissues and organs. These include bone tissue, skin, mucous membranes of internal organs, conjunctiva of the eye, vascular intima. The combination of the effect of laser treatment and the use of biological factors, including bone marrow stem cells, is very promising. However, though the mechanisms of this process have not yet been fully clarified, there is a convincing evidence of restoration of damaged tissues with different degrees of development of the pathological process. Optimal dosimetry of laser stimulation of other body tissues has not been studied. This requires further research which should be carried out in cooperation with medical doctors of various specialties, physicists engaged in the improvement of laser technology, biochemists, and morphologists. Comprehensive experimental and clinical research in this area is within the power of serious universities and research institutions working in this direction. Regenerative laser medicine will significantly shorten the treatment time for skeletal fractures, stop the development of the degenerative process of joints and internal organs, avoid many severe surgical interventions on the spine, joints, blood vessels and other tissues that currently require prosthetics. Its promising and striking results can be laser treatment of trophic ulcers and soft tissue pressure ulcers, as well as of ulcers of the esophagus, stomach, intestines, bladder, and other organs carried out endoscopically.

## 5. Conclusions

Summing up the results of many years of painstaking work of a large team of physicists, biologists, morphologists, orthopedists, and neurosurgeons who have studied the possibilities of regeneration of cartilaginous tissue—the most complex and rigid body structure, we can say with confidence that a significant contribution has been made to the regenerative medicine. The laser turned out to be a powerful stimulator of tissue regeneration. New, previously unknown parameters of a modifying non-destructive effect on tissues have proven to be very effective for the restoration of degeneratively altered intervertebral discs. These changes in the disc underlie DDD. Thus, by stimulating the process of IVD regeneration, it is possible to prevent the development of DDD. Based on numerous experimental and clinical studies, the optimal laser has been selected. It turned out to be a 1560 nm Er fiber laser. It penetrates sufficiently deep into tissues with adequate parameters (power, duration, and pulse repetition rate) upon exposure, it does not destroy tissues and helps to restore the nutrition of disc tissues, restoring pores in hyaline cartilage and stimulating the activity of chondrocytes and stem cells. As a result, the ruptures of the AF are quickly replaced by hyaline cartilage, which closes its defects, preventing the recurrence of disc herniation. In place of the destroyed NP, a new (first described) type of cartilage tissue is formed, fibrohyaline cartilage, which has both elasticity and high strength as well as resistance to stress. This newly formed cartilage significantly reduces the instability of the segment, which helps to reduce the stress on the joints, stopping the development of the degenerative process in the spine. Within 3–6 months, almost 90% of patients restore their quality of life, they can play sports and enjoy an active lifestyle without disease recurrence. Amazingly, this requires a one-time treatment procedure. At present, more than five thousand patients have been treated with LRD, including more than three thousand patients with spine DDD who had the puncture-type surgeries, Most of these patients are young, of active age and demonstrated excellent clinical results. MRI examinations make it possible to see and thus objectify the process. restoration of structural formations of the IVD and facet joints. The method of LRD allows us to reduce the likelihood of recurrence of disc herniation after microdiscectomy by several times (from 4

to 1%) and this percentage is practically zero during puncture or endoscopic removal of a disc herniation. In last 15 years, the combination of removal of a herniated disc with laser reconstruction has been used in more than two thousand patients.

Of course, this new method of treatment requires further studying and improvement, but now it has already begun to be used in the treatment of degenerative lesions of large joints. Preliminary results show its safety and efficacy, which makes it possible to use it in the early stages of osteoarthritis, which in the future will significantly reduce the number of prostheses for large joints.

**Author Contributions:** Conceptualization, A.B., A.S. and E.S.; theoretical investigations E.S.; experiential animal investigations A.B., V.B., I.A.B. and E.S.; clinical study, A.B., I.A.B. and V.B.; histological investigation, A.S.; writing—original draft preparation, A.B. and E.S. All authors have read and agreed to the published version of the manuscript.

**Funding:** This research received no external funding.

**Institutional Review Board Statement:** The Independent Interdisciplinary Committee on Ethical Experiments, Moscow, Russia approved the Clinical Study "Laser Reconstruction of Intervertebral Discs", Approval No. 13, 16 August 2019.

**Informed Consent Statement:** Informed consent was obtained from all subjects involved in the study.

**Acknowledgments:** The authors thank Arcuo Medical Inc., USA for financial assistance, for the development and provision of LRD equipment with a feedback control system. A.B. and the V.B. thanks IRE Polus, Russia for financial assistance and provision of a fiber laser. A.S. thanks the Ministry of Science and Higher Education of the Russian Federation within the framework of state support for the creation and development of World-Class Research Centers Digital biodesign and personalized healthcare No. 075-15-2020-926.

**Conflicts of Interest:** The authors declare no conflict of interest.

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
