# Peer review of "Laser Reconstruction of Spinal Discs Experiments and Clinic"

_applsci, doi:10.3390/app12020675_

Round 1

Reviewer 1 Report

The present review paper is well written and designed.

I sugesst to the authors to add a last paragraph (before conclusion) on the future perspective. I also suggest to add a paragraph on the use of laser treatment in in vitro model (for e.g. doi: 10.1007/s10103-017-2384-6;  doi: 10.1089/pho.2014.3864).

Author Response

(1) The last paragraph on future perspective has been added  before the conclusion.

(2) We have added a paragraph on the use of low-level laser therapy with relating references to section 2.3.

Reviewer 2 Report

The manuscript by Baskov et al., descibes a problem of DDD treatment. The topic could be of a high interest, but very high percent (aprox. 25%) of the references are self-citations! This is unacceptable and has to be improved before any further considerations!

Author Response

The list of references has been corrected. Seven self-citation references were removed, and seven new references were added.

Reviewer 3 Report

It seems like a well-written manuscript in general, but there are some critical errors and points that need to be addressed: 

  • There are lots of formatting issues. Sec 3.2 is all italics, there is a font discrepancy in Line 186, also there are numerous spacing errors throughout.
  • There are some grammatical problems, wrong choice of words etc.
  • Line 130 - 'careful selection of patients..' - I am curious on what basis this careful selection has been made?
  • Sec 3.1 - 'In vivo experimentation on the rabbits' : The authors have given a thorough explanation of how the ECM, AF, and NP transform with laser exposure and consequent healing, but I think a study on the effect on pain receptors would also be helpful, because as the authors highlighted themselves - there are two factors in DDD treatment : pain alleviation and the actual healing of the IVD.
  • I think the paper lacks much discussion on future scope and outlook, esp. where and how research should proceed from now on, etc.

Round 2

Reviewer 3 Report

The authors have addressed all the queries and have taken care of the revisions as mentioned. The manuscript can be accepted in its present form with few minor editorial modifications.